# Object-Centric Learning with Slot Attention

**Francesco Locatello**[2,3,†,*], **Dirk Weissenborn**[1], **Thomas Unterthiner**[1], **Aravindh Mahendran**[1],
**Georg Heigold**[1], **Jakob Uszkoreit**[1], **Alexey Dosovitskiy**[1,‡], and **Thomas Kipf**[1,‡,*]

[1]Google Research, Brain Team
[2]Dept. of Computer Science, ETH Zurich
[3]Max-Planck Institute for Intelligent Systems

## Abstract

Learning object-centric representations of complex scenes is a promising step towards enabling efficient abstract reasoning from low-level perceptual features. Yet, most deep learning approaches learn distributed representations that do not capture the compositional properties of natural scenes. In this paper, we present the Slot Attention module, an architectural component that interfaces with perceptual representations such as the output of a convolutional neural network and produces a set of task-dependent abstract representations which we call slots. These slots are exchangeable and can bind to any object in the input by specializing through a competitive procedure over multiple rounds of attention. We empirically demonstrate that Slot Attention can extract object-centric representations that enable generalization to unseen compositions when trained on unsupervised object discovery and supervised property prediction tasks.

## 1 Introduction

Object-centric representations have the potential to improve sample efficiency and generalization of machine learning algorithms across a range of application domains, such as visual reasoning [1], modeling of structured environments [2], multi-agent modeling [3–5], and simulation of interacting physical systems [6–8]. Obtaining object-centric representations from raw perceptual input, such as an image or a video, is challenging and often requires either supervision [1, 3, 9, 10] or task-specific architectures [2, 11]. As a result, the step of learning an object-centric representation is often skipped entirely. Instead, models are typically trained to operate on a structured representation of the environment that is obtained, for example, from the internal representation of a simulator [6, 8] or of a game engine [4, 5].

To overcome this challenge, we introduce the Slot Attention module, a differentiable *interface* between perceptual representations (e.g., the output of a CNN) and a *set* of variables called *slots*. Using an iterative attention mechanism, Slot Attention produces a set of output vectors with permutation symmetry. Unlike *capsules* used in Capsule Networks [12, 13], slots produced by Slot Attention do not specialize to one particular type or class of object, which could harm generalization. Instead, they act akin to *object files* [14], i.e., slots use a common representational format: each slot can store (and bind to) any object in the input. This allows Slot Attention to generalize in a systematic way to unseen compositions, more objects, and more slots.

Slot Attention is a simple and easy to implement architectural component that can be placed, for example, on top of a CNN [15] encoder to extract object representations from an image and is trained end-to-end with a downstream task. In this paper, we consider image reconstruction and set prediction as downstream tasks to showcase the versatility of our module both in a challenging unsupervised object discovery setup and in a supervised task involving set-structured object property prediction.

---

[†]Work done while interning at Google, [*]equal contribution, [‡]equal advising. Contact: `tkipf@google.com`

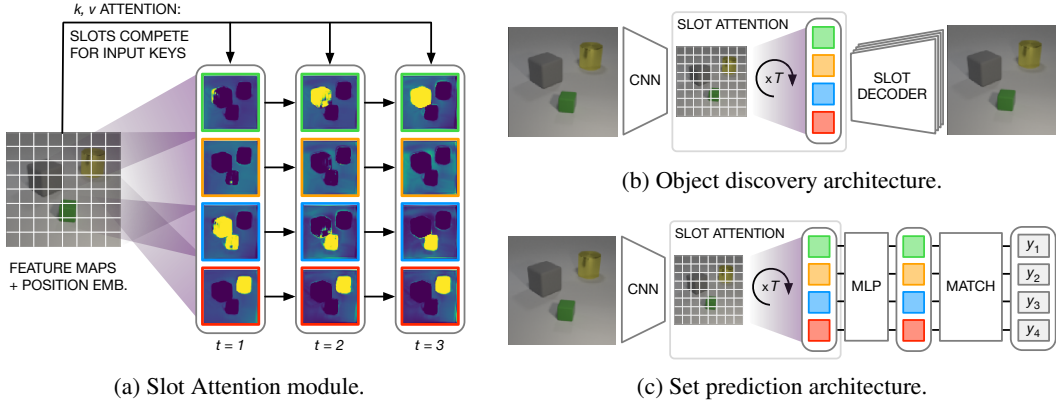

(a) Slot Attention module.

(b) Object discovery architecture.

(c) Set prediction architecture.

Figure 1: (**a**) Slot Attention module and example applications to (**b**) unsupervised object discovery and (**c**) supervised set prediction with labeled targets $y_i$. See main text for details.

**Our main contributions** are as follows: (i) We introduce the Slot Attention module, a simple architectural component at the interface between perceptual representations (such as the output of a CNN) and representations structured as a set. (ii) We apply a Slot Attention-based architecture to unsupervised object discovery, where it matches or outperforms relevant state-of-the-art approaches [16, 17], while being more memory efficient and significantly faster to train. (iii) We demonstrate that the Slot Attention module can be used for supervised object property prediction, where the attention mechanism learns to highlight individual objects without receiving direct supervision on object segmentation.

## 2   Methods

In this section, we introduce the Slot Attention module (Figure 1a; Section 2.1) and demonstrate how it can be integrated into an architecture for unsupervised object discovery (Figure 1b; Section 2.2) and into a set prediction architecture (Figure 1c; Section 2.3).

### 2.1   Slot Attention Module

The Slot Attention module (Figure 1a) maps from a set of $N$ input feature vectors to a set of $K$ output vectors that we refer to as *slots*. Each vector in this output set can, for example, describe an object or an entity in the input. The overall module is described in Algorithm 1 in pseudo-code[1].

Slot Attention uses an iterative attention mechanism to map from its inputs to the slots. Slots are initialized at random and thereafter refined at each iteration $t = 1 \dots T$ to bind to a particular part (or grouping) of the input features. Randomly sampling initial slot representations from a common distribution allows Slot Attention to generalize to a different number of slots at test time.

At each iteration, slots *compete* for explaining parts of the input via a softmax-based attention mechanism [18–20] and update their representation using a recurrent update function. The final representation in each slot can be used in downstream tasks such as unsupervised object discovery (Figure 1b) or supervised set prediction (Figure 1c).

We now describe a single iteration of Slot Attention on a set of input features, $\texttt{inputs} \in \mathbb{R}^{N \times D_{\texttt{inputs}}}$, with $K$ output slots of dimension $D_{\texttt{slots}}$ (we omit the batch dimension for clarity). We use learnable linear transformations $k$, $q$, and $v$ to map inputs and slots to a common dimension $D$.

Slot Attention uses dot-product attention [19] with attention coefficients that are normalized over the slots, i.e., the queries of the attention mechanism. This choice of normalization introduces competition between the slots for explaining parts of the input.

**Algorithm 1** Slot Attention module. The input is a set of $N$ vectors of dimension $D_{\texttt{inputs}}$ which is mapped to a set of $K$ slots of dimension $D_{\texttt{slots}}$. We initialize the slots by sampling their initial values as independent samples from a Gaussian distribution with shared, learnable parameters $\mu \in \mathbb{R}^{D_{\texttt{slots}}}$ and $\sigma \in \mathbb{R}^{D_{\texttt{slots}}}$. In our experiments we set the number of iterations to $T = 3$.

1: **Input**: $\texttt{inputs} \in \mathbb{R}^{N \times D_{\texttt{inputs}}}$, $\texttt{slots} \sim \mathcal{N}(\mu,\, \mathrm{diag}(\sigma)) \in \mathbb{R}^{K \times D_{\texttt{slots}}}$
2: **Layer params**: $k$, $q$, $v$: linear projections for attention; GRU; MLP; LayerNorm(x3)
3:    $\texttt{inputs} = \texttt{LayerNorm}(\texttt{inputs})$
4:    **for** $t = 0 \dots T$
5:      $\texttt{slots\_prev} = \texttt{slots}$
6:      $\texttt{slots} = \texttt{LayerNorm}(\texttt{slots})$
7:      $\texttt{attn} = \texttt{Softmax}\left(\frac{1}{\sqrt{D}}k(\texttt{inputs}) \cdot q(\texttt{slots})^T,\ \texttt{axis='slots'}\right)$     \# norm. over slots
8:      $\texttt{updates} = \texttt{WeightedMean}(\texttt{weights=attn} + \epsilon,\ \texttt{values=}v(\texttt{inputs}))$     \# aggregate
9:      $\texttt{slots} = \texttt{GRU}(\texttt{state=slots\_prev},\ \texttt{inputs=updates})$     \# GRU update (per slot)
10:     $\texttt{slots} \mathrel{+}= \texttt{MLP}(\texttt{LayerNorm}(\texttt{slots}))$     \# optional residual MLP (per slot)
11:   **return** $\texttt{slots}$

We further follow the common practice of setting the softmax temperature to a fixed value of $\sqrt{D}$ [20]:

$$\texttt{attn}_{i,j} := \frac{e^{M_{i,j}}}{\sum_l e^{M_{i,l}}} \qquad \text{where} \qquad M := \frac{1}{\sqrt{D}}k(\texttt{inputs}) \cdot q(\texttt{slots})^T \in \mathbb{R}^{N \times K}. \qquad (1)$$

In other words, the normalization ensures that attention coefficients sum to one for each individual input feature vector, which prevents the attention mechanism from ignoring parts of the input. To aggregate the input values to their assigned slots, we use a weighted mean as follows:

$$\texttt{updates} := W^T \cdot v(\texttt{inputs}) \in \mathbb{R}^{K \times D} \qquad \text{where} \qquad W_{i,j} := \frac{\texttt{attn}_{i,j}}{\sum_{l=1}^{N} \texttt{attn}_{l,j}}. \qquad (2)$$

The weighted mean helps improve stability of the attention mechanism (compared to using a weighted sum) as in our case the attention coefficients are normalized over the slots. In practice we further add a small offset $\epsilon$ to the attention coefficients to avoid numerical instability.

The aggregated $\texttt{updates}$ are finally used to update the slots via a learned recurrent function, for which we use a Gated Recurrent Unit (GRU) [21] with $D_{\texttt{slots}}$ hidden units. We found that transforming the GRU output with an (optional) multi-layer perceptron (MLP) with ReLU activation and a residual connection [22] can help improve performance. Both the GRU and the residual MLP are applied independently on each slot with shared parameters. We apply layer normalization (LayerNorm) [23] both to the inputs of the module and to the slot features at the beginning of each iteration and before applying the residual MLP. While this is not strictly necessary, we found that it helps speed up training convergence. The overall time-complexity of the module is $\mathcal{O}(T \cdot D \cdot N \cdot K)$.

We identify two key properties of Slot Attention: (1) permutation invariance with respect to the input (i.e., the output is independent of permutations applied to the input and hence suitable for sets) and (2) permutation equivariance with respect to the order of the slots (i.e., permuting the order of the slots after their initialization is equivalent to permuting the output of the module). More formally:

**Proposition 1.** *Let* SlotAttention$(\texttt{inputs}, \texttt{slots}) \in \mathbb{R}^{K \times D_{\texttt{slots}}}$ *be the output of the Slot Attention module (Algorithm 1), where* $\texttt{inputs} \in \mathbb{R}^{N \times D_{\texttt{inputs}}}$ *and* $\texttt{slots} \in \mathbb{R}^{K \times D_{\texttt{slots}}}$. *Let* $\pi_i \in \mathbb{R}^{N \times N}$ *and* $\pi_s \in \mathbb{R}^{K \times K}$ *be arbitrary permutation matrices. Then, the following holds:*

$$\text{SlotAttention}(\pi_i \cdot \texttt{inputs}, \pi_s \cdot \texttt{slots}) = \pi_s \cdot \text{SlotAttention}(\texttt{inputs}, \texttt{slots}).$$

The proof is in the supplementary material. The permutation equivariance property is important to ensure that slots learn a common representational format and that each slot can bind to any object in the input.

## 2.2 Object Discovery

Set-structured hidden representations are an attractive choice for learning about objects in an unsupervised fashion: each set element can capture the properties of an object in a scene, without assuming

a particular order in which objects are described. Since Slot Attention transforms input representations into a set of vectors, it can be used as part of the encoder in an autoencoder architecture for unsupervised object discovery. The autoencoder is tasked to encode an image into a set of hidden representations (i.e., slots) that, taken together, can be decoded back into the image space to reconstruct the original input. The slots thereby act as a representational bottleneck and the architecture of the decoder (or decoding process) is typically chosen such that each slot decodes only a region or part of the image [16, 17, 24–27]. These regions/parts are then combined to arrive at the full reconstructed image.

**Encoder** Our encoder consists of two components: (i) a CNN backbone augmented with positional embeddings, followed by (ii) a Slot Attention module. The output of Slot Attention is a set of slots, that represent a grouping of the scene (e.g. in terms of objects).

**Decoder** Each slot is decoded individually with the help of a spatial broadcast decoder [28], as used in IODINE [16]: slot representations are broadcasted onto a 2D grid (per slot) and augmented with position embeddings. Each such grid is decoded using a CNN (with parameters shared across the slots) to produce an output of size $W \times H \times 4$, where $W$ and $H$ are width and height of the image, respectively. The output channels encode RGB color channels and an (unnormalized) alpha mask. We subsequently normalize the alpha masks across slots using a `Softmax` and use them as mixture weights to combine the individual reconstructions into a single RGB image.

## 2.3 Set Prediction

Set representations are commonly used in tasks across many data modalities ranging from point cloud prediction [29, 30], classifying multiple objects in an image [31], or generation of molecules with desired properties [32, 33]. In the example considered in this paper, we are given an input image and a set of prediction targets, each describing an object in the scene. The key challenge in predicting sets is that there are $K!$ possible equivalent representations for a set of $K$ elements, as the order of the targets is arbitrary. This inductive bias needs to be explicitly modeled in the architecture to avoid discontinuities in the learning process, e.g. when two semantically specialized slots swap their content throughout training [31, 34]. The output order of Slot Attention is random and independent of the input order, which addresses this issue. Therefore, Slot Attention can be used to turn a distributed representation of an input scene into a set representation where each object can be separately classified with a standard classifier as shown in Figure 1c.

**Encoder** We use the same encoder architecture as in the object discovery setting (Section 2.2), namely a CNN backbone augmented with positional embeddings, followed by Slot Attention, to arrive at a set of slot representations.

**Classifier** For each slot, we apply a MLP with parameters shared between slots. As the order of both predictions and labels is arbitrary, we match them using the Hungarian algorithm [35]. We leave the exploration of other matching algorithms [36, 37] for future work.

## 3 Related Work

**Object discovery** Our object discovery architecture is closely related to a line of recent work on compositional generative scene models [16, 17, 24–27, 38–44] that represent a scene in terms of a collection of latent variables with the same representational format. Closest to our approach is the IODINE [16] model, which uses iterative variational inference [45] to infer a set of latent variables, each describing an object in an image. In each inference iteration, IODINE performs a decoding step followed by a comparison in pixel space and a subsequent encoding step. Related models such as MONet [17] and GENESIS [27] similarly use multiple encode-decode steps. Our model instead replaces this procedure with a single encoding step using iterated attention, which improves computational efficiency. Further, this allows our architecture to infer object representations and attention masks even in the absence of a decoder, opening up extensions beyond auto-encoding, such as contrastive representation learning for object discovery [46] or direct optimization of a downstream task like control or planning. Our attention-based routing procedure could also be employed in conjunction with patch-based decoders, used in architectures such as AIR [26], SQAIR [40], and related approaches [41–44], as an alternative to the typically employed autoregressive encoder [26, 40]. Our approach is orthogonal to methods using adversarial training [47–49] or contrastive learning [46] for object discovery: utilizing Slot Attention in such a setting is an interesting avenue for future work.

**Neural networks for sets** A range of recent methods explore set encoding [34, 50, 51], generation [31, 52], and set-to-set mappings [20, 53]. Graph neural networks [54–57] and in particular the self-attention mechanism of the Transformer model [20] are frequently used to transform sets of elements with constant cardinality (i.e., number of set elements). Slot Attention addresses the problem of mapping from one set to another set of different cardinality while respecting permutation symmetry of both the input and the output set. The Deep Set Prediction Network (DSPN) [31, 58] respects permutation symmetry by running an inner gradient descent loop for each example, which requires many steps for convergence and careful tuning of several loss hyperparmeters. Instead, Slot Attention directly maps from set to set using only a few attention iterations and a single task-specific loss function. In concurrent work, both the DETR [59] and the TSPN [60] model propose to use a Transformer [20] for conditional set generation. Most related approaches, including DiffPool [61], Set Transformers [53], DSPN [31], and DETR [59] use a learned per-element initialization (i.e., separate parameters for each set element), which prevents these approaches from generalizing to more set elements at test time.

**Iterative routing** Our iterative attention mechanism shares similarlities with iterative *routing* mechanisms typically employed in variants of Capsule Networks [12, 13, 62]. The closest such variant is inverted dot-product attention routing [62] which similarly uses a dot product attention mechanism to obtain assignment coefficients between representations. Their method (in line with other capsule models) however does not have permutation symmetry as each input-output pair is assigned a separately parameterized transformation. The low-level details in how the attention mechanism is normalized and how updates are aggregated, and the considered applications also differ significantly between the two approaches.

**Interacting memory models** Slot Attention can be seen as a variant of interacting memory models [9, 39, 46, 63–68], which utilize a set of slots and their pairwise interactions to reason about elements in the input (e.g. objects in a video). Common components of these models are (i) a recurrent update function that acts independently on individual slots and (ii) an interaction function that introduces communication between slots. Typically, slots in these models are fully symmetric with shared recurrent update functions and interaction functions for all slots, with the exception of the RIM model [67], which uses a separate set of parameters for each slot. Notably, RMC [63] and RIM [67] introduce an attention mechanism to aggregate information from inputs to slots. In Slot Attention, the attention-based assignment from inputs to slots is normalized over the slots (as opposed to solely over the inputs), which introduces competition between the slots to perform a clustering of the input. Further, we do not consider temporal data in this work and instead use the recurrent update function to iteratively refine predictions for a single, static input.

**Mixtures of experts** Expert models [67, 69–72] are related to our slot-based approach, but do not fully share parameters between individual experts. This results in the specialization of individual experts to, e.g., different tasks or object types. In Slot Attention, slots use a common representational format and each slot can bind to any part of the input.

**Soft clustering** Our routing procedure is related to soft k-means clustering [73] (where slots corresponds to cluster centroids) with two key differences: We use a dot product similarity with learned linear projections and we use a parameterized, learnable update function. Variants of soft k-means clustering with learnable, cluster-specific parameters have been introduced in the computer vision [74] and speech recognition communities [75], but they differ from our approach in that they do not use a recurrent, multi-step update, and do not respect permutation symmetry (cluster centers act as a fixed, ordered dictionary after training). The inducing point mechanism of the Set Transformer [53] and the image-to-slot attention mechanism in DETR [59] can be seen as extensions of these ordered, single-step approaches using multiple attention heads (i.e., multiple similarity functions) for each cluster assignment.

**Recurrent attention** Our method is related to recurrent attention models used in image modeling and scene decomposition [26, 40, 76–78], and for set prediction [79]. Recurrent models for set prediction have also been considered in this context without using attention mechanisms [80, 81]. This line of work frequently uses permutation-invariant loss functions [79, 80, 82], but relies on inferring one slot, representation, or label per time step in an auto-regressive manner, whereas Slot Attention updates all slots simultaneously at each step, hence fully respecting permutation symmetry.

|  | CLEVR6 | Multi-dSprites | Tetrominoes |
|---|---|---|---|
| Slot Attention | $98.8 \pm 0.3$ | $91.3 \pm 0.3$ | $99.5 \pm 0.2$* |
| IODINE [16] | $98.8 \pm 0.0$ | $76.7 \pm 5.6$ | $99.2 \pm 0.4$ |
| MONet [17] | $96.2 \pm 0.6$ | $90.4 \pm 0.8$ | — |
| Slot MLP | $60.4 \pm 6.6$ | $60.3 \pm 1.8$ | $25.1 \pm 34.3$ |

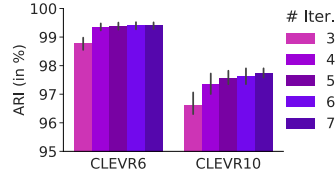

Table 1 & Figure 2: (**Left**) Adjusted Rand Index (ARI) scores (in %, mean $\pm$ stddev for 5 seeds) for unsupervised object discovery in multi-object datasets. In line with previous works [16, 17, 27], we exclude background labels in ARI evaluation. *denotes that one outlier was excluded from evaluation. (**Right**) Effect of increasing the number of Slot Attention iterations $T$ at test time (for a model trained on CLEVR6 with $T = 3$ and $K = 7$ slots), tested on CLEVR6 ($K = 7$) and CLEVR10 ($K = 11$).

## 4 Experiments

The goal of this section is to evaluate the Slot Attention module on two object-centric tasks—one being supervised and the other one being unsupervised—as described in Sections 2.2 and 2.3. We compare against specialized state-of-the-art methods [16, 17, 31] for each respective task. We provide further details on experiments and implementation, and additional qualitative results and ablation studies in the supplementary material.

**Baselines** In the unsupervised object discovery experiments, we compare against two recent state-of-the-art models: IODINE [16] and MONet [17]. For supervised object property prediction, we compare against Deep Set Prediction Networks (DSPN) [31]. DSPN is the only set prediction model that respects permutation symmetry that we are aware of, other than our proposed model. In both tasks, we further compare against a simple MLP-based baseline that we term Slot MLP. This model replaces Slot Attention with an MLP that maps from the CNN feature maps (resized and flattened) to the (now ordered) slot representation. For the MONet, IODINE, and DSPN baselines, we compare with the published numbers in [16, 31] as we use the same experimental setup.

**Datasets** For the object discovery experiments, we use the following three multi-object datasets [83]: CLEVR (with masks), Multi-dSprites, and Tetrominoes. CLEVR (with masks) is a version of the CLEVR dataset with segmentation mask annotations. Similar to IODINE [16], we only use the first 70K samples from the CLEVR (with masks) dataset for training and we crop images to highlight objects in the center. For Multi-dSprites and Tetrominoes, we use the first 60K samples. As in [16], we evaluate on 320 test examples for object discovery. For set prediction, we use the original CLEVR dataset [84] which contains a training-validation split of 70K and 15K images of rendered objects respectively. Each image can contain between three and ten objects and has property annotations for each object (position, shape, material, color, and size). In some experiments, we filter the CLEVR dataset to contain only scenes with at maximum 6 objects; we call this dataset CLEVR6 and we refer to the original full dataset as CLEVR10 for clarity.

### 4.1 Object Discovery

**Training** The training setup is unsupervised: the learning signal is provided by the (mean squared) image reconstruction error. We train the model using the Adam optimizer [85] with a learning rate of $4 \times 10^{-4}$ and a batch size of 64 (using a single GPU). We further make use of learning rate warmup [86] to prevent early saturation of the attention mechanism and an exponential decay schedule in the learning rate, which we found to reduce variance. At training time, we use $T = 3$ iterations of Slot Attention. We use the same training setting across all datasets, apart from the number of slots $K$: we use $K = 7$ slots for CLEVR6, $K = 6$ slots for Multi-dSprites (max. 5 objects per scene), and $K = 4$ for Tetrominoes (3 objects per scene). Even though the number of slots in Slot Attention can be set to a different value for each input example, we use the same value $K$ for all examples in the training set to allow for easier batching.

**Metrics** In line with previous works [16, 17], we compare the alpha masks produced by the decoder (for each individual object slot) with the ground truth segmentation (excluding the background) using the Adjusted Rand Index (ARI) score [87, 88]. ARI is a score to measure clustering similarity, ranging from 0 (random) to 1 (perfect match). To compute the ARI score, we use the implementation provided by Kabra et al. [83].

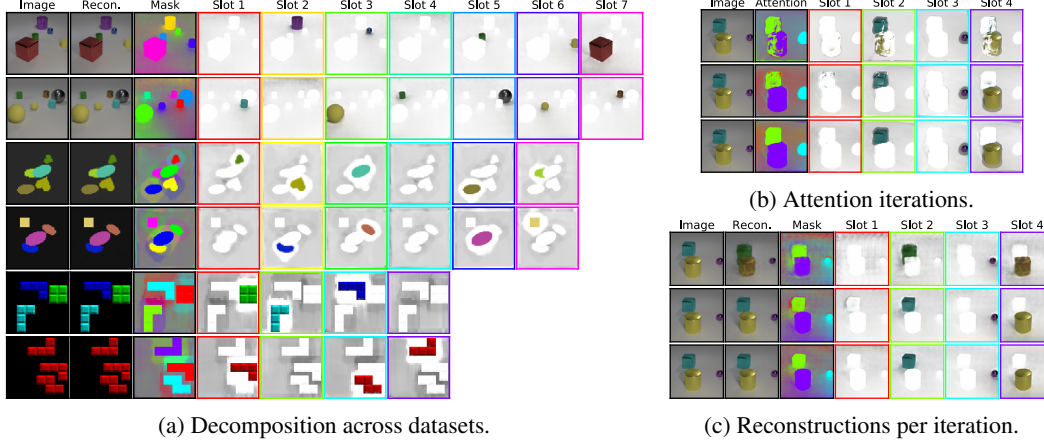

(b) Attention iterations.

(a) Decomposition across datasets.

(c) Reconstructions per iteration.

Figure 3: (**a**) Visualization of per-slot reconstructions and alpha masks in the unsupervised training setting (object discovery). Top rows: CLEVR6, middle rows: Multi-dSprites, bottom rows: Tetrominoes. (**b**) Attention masks (`attn`) for each iteration, only using four object slots at test time on CLEVR6. (**c**) Per-iteration reconstructions and reconstruction masks (from decoder). Border colors for slots correspond to colors of segmentation masks used in the combined mask visualization (third column). We visualize individual slot reconstructions multiplied with their respective alpha mask, using the visualization script from [16].

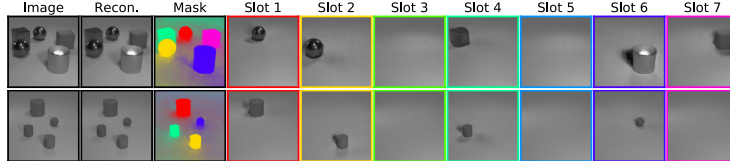

Figure 4: Visualization of (per-slot) reconstructions and masks of a Slot Attention model trained on a greyscale version of CLEVR6, where it achieves $98.5 \pm 0.3\%$ ARI. Here, we show the full reconstruction of each slot (i.e., without multiplication with their respective alpha mask).

**Results** Quantitative results are summarized in Table 1 and Figure 2. In general, we observe that our model compares favorably against two recent state-of-the-art baselines: IODINE [16] and MONet [17]. We also compare against a simple MLP-based baseline (Slot MLP) which performs better than chance, but due to its ordered representation is unable to model the compositional nature of this task. We note a failure mode of our model: In rare cases it can get stuck in a suboptimal solution on the Tetrominoes dataset, where it segments the image into stripes. This leads to a significantly higher reconstruction error on the training set, and hence such an outlier can easily be identified at training time. We excluded a single such outlier (1 out of 5 seeds) from the final score in Table 1. We expect that careful tuning of the training hyperparameters particularly for this dataset could alleviate this issue, but we opted for a single setting shared across all datasets for simplicity.

Compared to IODINE [16], Slot Attention is significantly more efficient in terms of both memory consumption and runtime. On CLEVR6, we can use a batch size of up to 64 on a single V100 GPU with 16GB of RAM as opposed to 4 in [16] using the same type of hardware. Similarly, when using 8 V100 GPUs in parallel, model training on CLEVR6 takes approximately 24hrs for Slot Attention as opposed to approximately 7 days for IODINE [16].

In Figure 2, we investigate to what degree our model generalizes when using more Slot Attention iterations at test time, while being trained with a fixed number of $T = 3$ iterations. We further evaluate generalization to more objects (CLEVR10) compared to the training set (CLEVR6). We observe that segmentation scores significantly improve beyond the numbers reported in Table 1 when using more iterations. This improvement is stronger when testing on CLEVR10 scenes with more objects. For this experiment, we increase the number of slots from $K = 7$ (training) to $K = 11$ at test time. Overall, segmentation performance remains strong even when testing on scenes that contain more objects than seen during training.

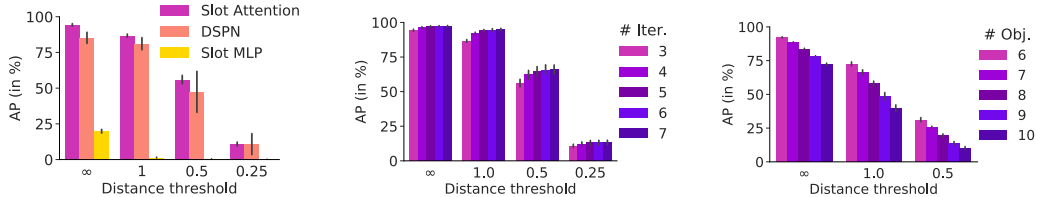

Figure 5: (**Left**) AP at different distance thresholds on CLEVR10 (with $K = 10$). (**Center**) AP for the Slot Attention model with different number of iterations. The models are trained with 3 iterations and tested with iterations ranging from 3 to 7. (**Right**) AP for Slot Attention trained on CLEVR6 ($K = 6$) and tested on scenes containing exactly $N$ objects (with $N = K$ from 6 to 10).

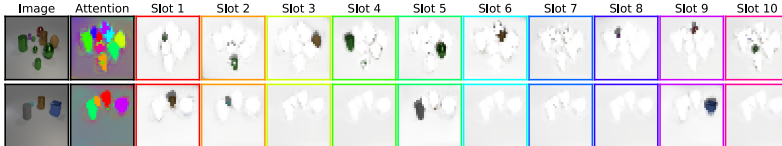

Figure 6: Visualization of the attention masks on CLEVR10 for two examples with 9 and 4 objects, respectively, for a model trained on the property prediction task. The masks are upsampled to $128 \times 128$ for this visualization to match the resolution of input image.

We visualize discovered object segmentations in Figure 3 for all three datasets. The model learns to keep slots empty (only capturing the background) if there are more slots than objects. We find that Slot Attention typically spreads the uniform background across all slots instead of capturing it in just a single slot, which is likely an artifact of the attention mechanism that does not harm object disentanglement or reconstruction quality. We further visualize how the attention mechanism segments the scene over the individual attention iterations, and we inspect scene reconstructions from each individual iteration (the model has been trained to reconstruct only after the final iteration). It can be seen that the attention mechanism learns to specialize on the extraction of individual objects already at the second iteration, whereas the attention map of the first iteration still maps parts of multiple objects into a single slot.

To evaluate whether Slot Attention can perform segmentation without relying on color cues, we further run experiments on a binarized version of multi-dSprites with white objects on black background, and on a greyscale version of CLEVR6. We use the binarized multi-dSprites dataset from Kabra et al. [83], for which Slot Attention achieves $69.4 \pm 0.9\%$ ARI using $K = 4$ slots, compared to $64.8 \pm 17.2\%$ for IODINE [16] and $68.5 \pm 1.7\%$ for R-NEM [39], as reported in [16]. Slot Attention performs competitively in decomposing scenes into objects based on shape cues only. We visualize discovered object segmentations for the Slot Attention model trained on greyscale CLEVR6 in Figure 4, which Slot Attention handles without issue despite the lack of object color as a distinguishing feature.

As our object discovery architecture uses the same decoder and reconstruction loss as IODINE [16], we expect it to similarly struggle with scenes containing more complicated backgrounds and textures. Utilizing different perceptual [49, 89] or contrastive losses [46] could help overcome this limitation. We discuss further limitations and future work in Section 5 and in the supplementary material.

**Summary** Slot Attention is highly competitive with prior approaches on unsupervised scene decomposition, both in terms of quality of object segmentation and in terms of training speed and memory efficiency. At test time, Slot Attention can be used without a decoder to obtain object-centric representations from an unseen scene.

## 4.2 Set Prediction

**Training** We train our model using the same hyperparameters as in Section 4.1 except we use a batch size of 512 and striding in the encoder. On CLEVR10, we use $K = 10$ object slots to be in line with [31]. The Slot Attention model is trained using a single NVIDIA Tesla V100 GPU with 16GB of RAM.

**Metrics** Following Zhang et al. [31], we compute the Average Precision (AP) as commonly used in object detection [90]. A prediction (object properties and position) is considered correct if there is a matching object with exactly the same properties (shape, material, color, and size) within a certain distance threshold ($\infty$ means we do not enforce any threshold). The predicted position coordinates are scaled to $[-3, 3]$. We zero-pad the targets and predict an additional indicator score in $[0, 1]$ corresponding to the presence probability of an object (1 means there is an object) which we then use as prediction confidence to compute the AP.

**Results** In Figure 5 (left) we report results in terms of Average Precision for supervised object property prediction on CLEVR10 (using $T = 3$ for Slot Attention at both train and test time). We compare to both the DSPN results of [31] and the Slot MLP baseline. Overall, we observe that our approach matches or outperforms the DSPN baseline. The performance of our method degrades gracefully at more challenging distance thresholds (for the object position feature) maintaining a reasonably small variance. Note that the DSPN baseline [31] uses a significantly deeper ResNet 34 [22] image encoder. In Figure 5 (center) we observe that increasing the number of attention iterations at test time generally improves performance. Slot Attention can naturally handle more objects at test time by changing the number of slots. In Figure 5 (right) we observe that the AP degrades gracefully if we train a model on CLEVR6 (with $K = 6$ slots) and test it with more objects.

Intuitively, to solve this set prediction task each slot should attend to a different object. In Figure 6, we visualize the attention maps of each slot for two CLEVR images. In general, we observe that the attention maps naturally segment the objects. We remark that the method is only trained to predict the property of the objects, without any segmentation mask. Quantitatively, we can evaluate the Adjusted Rand Index (ARI) scores of the attention masks. On CLEVR10 (with masks), the attention masks produced by Slot Attention achieve an ARI of $78.0\% \pm 2.9$ (to compute the ARI we downscale the input image to $32 \times 32$). Note that the masks evaluated in Table 1 are not the attention maps but are predicted by the object discovery decoder.

**Summary** Slot Attention learns a representation of objects for set-structured property prediction tasks and achieves results competitive with a prior state-of-the-art approach while being significantly easier to implement and tune. Further, the attention masks naturally segment the scene, which can be valuable for debugging and interpreting the predictions of the model.

# 5 Conclusion

We have presented the Slot Attention module, a versatile architectural component that learns object-centric abstract representations from low-level perceptual input. The iterative attention mechanism used in Slot Attention allows our model to learn a grouping strategy to decompose input features into a set of slot representations. In experiments on unsupervised visual scene decomposition and supervised object property prediction we have shown that Slot Attention is highly competitive with prior related approaches, while being more efficient in terms of memory consumption and computation.

A natural next step is to apply Slot Attention to video data or to other data modalities, e.g. for clustering of nodes in graphs, on top of a point cloud processing backbone or for textual or speech data. It is also promising to investigate other downstream tasks, such as reward prediction, visual reasoning, control, or planning.

# Broader Impact

The Slot Attention module allows to learn object-centric representations from perceptual input. As such, it is a general module that can be used in a wide range of domains and applications. In our paper, we only consider artificially generated datasets under well-controlled settings where slots are expected to specialize to objects. However, the specialization of our model is implicit and fully driven by the downstream task. We remark that as a concrete measure to assess whether the module specialized in unwanted ways, one can visualize the attention masks to understand how the input features are distributed across the slots (see Figure 6). While more work is required to properly address the usefulness of the attention coefficients in explaining the overall predictions of the network (especially if the input features are not human interpretable), we argue that they may serve as a step towards more transparent and interpretable predictions.

## Acknowledgements and Disclosure of Funding

**Acknowledgements**   We would like to thank Nal Kalchbrenner for general advise and feedback on the paper, Mostafa Dehghani, Klaus Greff, Bernhard Schölkopf, Klaus-Robert Müller, Adam Kosiorek, and Peter Battaglia for helpful discussions, and Rishabh Kabra for advise regarding the DeepMind Multi-Object Datasets.

**Disclosure of funding**   This work was done when Francesco Locatello was an intern at Google; Dirk Weissenborn, Thomas Unterthiner, Aravindh Mahendran, Georg Heigold, Jakob Uszkoreit, Alexey Dosovitskiy, and Thomas Kipf are employees at Google.

## Footnotes

[1]An implementation of Slot Attention is available at: `https://github.com/google-research/google-research/tree/master/slot_attention`.

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
