[Supplementary Material]

# Supplementary Material for
# Object-Centric Learning with Slot Attention

**Francesco Locatello**[2,3,†,*]**, Dirk Weissenborn**[1]**, Thomas Unterthiner**[1]**, Aravindh Mahendran**[1]**,
Georg Heigold**[1]**, Jakob Uszkoreit**[1]**, Alexey Dosovitskiy**[1,‡]**, and Thomas Kipf**[1,‡,*]

[1]Google Research, Brain Team
[2]Dept. of Computer Science, ETH Zurich
[3]Max-Planck Institute for Intelligent Systems

In Section 1, we highlight some limitations of our work as well as potential directions for future work. In Section 2, we report results of an ablation study on Slot Attention. In Section 3, we report further qualitative and quantitative results. In Section 4, we give the proof for Proposition 1. In Section 5, we report details on our implementation and experimental setting.

## 1   Limitations

We highlight several limitations of the Slot Attention module that could potentially be addressed in future work:

**Background treatment**  The background of a scene receives no special treatment in Slot Attention as all slots use the same representational format. Providing special treatment for backgrounds (e.g., by assigning a separate background slot) is interesting for future work.

**Translation symmetry**  The positional encoding used in our experiments is absolute and hence our module is not equivariant to translations. Using a patch-based object extraction process as in [1] or an attention mechanism with relative positional encoding [2] are promising extensions.

**Type of clustering**  Slot Attention does not know about objects per-se: segmentation is solely driven by the downstream task, i.e., Slot Attention does not distinguish between clustering objects, colors, or simply spatial regions, but relies on the downstream task to drive the specialization to objects.

**Communication between slots**  In Slot Attention, slots only communicate via the softmax attention over the input keys, which is normalized across slots. This choice of normalization establishes competition between the slots for explaining parts of the input, which can drive specialization of slots to objects. In some scenarios it can make sense to introduce a more explicit form of communication between slots, e.g. via slot-to-slot message passing in the form of a graph neural network as in [3, 4] or self-attention [5–9]. This can be beneficial for modeling systems of objects that interact dynamically [3, 4, 8] or for set generation conditioned on a single vector (as opposed to an image or a set of vectors) [9].

## 2   Model Ablations

In this section, we investigate the importance of individual components and modeling choices in the Slot Attention module and compare our default choice to a variety of reasonable alternatives. For simplicity, we report all results on a smaller validation set consisting of 500 validation images for property prediction (instead of 15K) and on 320 training images for object discovery. In the unsupervised case, results on the training set and on held-out validation examples are nearly identical.

---

[†]Work done while interning at Google, [*]equal contribution, [‡]equal advising. Contact: `tkipf@google.com`

**Value aggregation** In Figure 1, we show the effect of taking a weighted sum as opposed to a weighted average in Line 8 of Algorithm 1. The average stabilizes training and yields significantly higher ARI and Average Precision scores (especially at the more strict distance thresholds). We can obtain a similar effect by replacing the weighted mean with a weighted sum followed by layer normalization (LayerNorm) [10].

**Position embedding** In Figure 2, we observe that the position embedding is not necessary for predicting categorical object properties. However, the performance in predicting the object position clearly decreases if we remove the position embedding. Similarly, the ARI score in unsupervised object discovery is significantly lower when not adding positional information to the CNN feature maps before the Slot Attention module.

**Slot initialization** In Figure 3, we show the effect of learning a separate set of Gaussian mean and variance parameters for the initialization of each slot compared to the default setting of using a shared set of parameters for all slots. We observe that a per-slot parameterization can increase performance slightly for the supervised task, but decreases performance on the unsupervised task, compared to the default shared parameterization. We remark that when learning a separate set of parameters for each slot, adding additional slots at test time is not possible without re-training.

**Attention normalization axis** In Figure 4, we highlight the role of the softmax axis in the attention mechanism, i.e., over which dimension the normalization is performed. Taking the softmax over the slot axis induces competition among the slots for explaining parts of the input. When we take the softmax over the input axis instead[1] (as done in regular self-attention), the attention coefficients for each slot will be independent of all other slots, and hence slots have no means of exchanging information, which significantly harms performance on both tasks.

**Recurrent update function** In Figure 5, we highlight the role of the GRU in learning the update function for the slots as opposed to simply taking the output of Line 8 as the next value for the slots. We observe that the learned update function yields a noticable improvement.

**Attention iterations** In Figure 6, we show the impact of the number of attention iterations while training. We observe a clear benefit in having more than a single attention iteration. Having more than 3 attention iterations significantly slows down training convergence, which results in lower performance when trained for the same number of steps. This can likely be mitigated by instead decoding and applying a loss at every attention iteration as opposed to only the last one. We note that at test time, using more than 3 attention iterations (even when trained with only 3 iterations) generally improves performance.

**Layer normalization** In Figure 7, we show that applying layer normalization (LayerNorm) [10] to the inputs and to the slot representations at each iteration in the Slot Attention module improves predictive performance. For set prediction, it particularly improves its ability to predict position accurately, likely because it leads to faster convergence at training time.

**Feedforward network** In Figure 8, we show that the residual MLP after the GRU is optional and may slow down convergence in property prediction, but may slightly improve performance on object discovery.

**Softmax temperature** In Figure 9, we show the effect of the softmax temperature. The scaling of $\sqrt{D}$ clearly improves the performance on both tasks.

**Offset for numerical stability** In Figure 10, we show that adding a small offset to the attention maps (for numerical stability) as in Algorithm 1 compared to the alternative of adding an offset to the denominator in the weighted mean does not significantly change the result in either task.

**Learning rate schedules** In Figure 11 and 12, we show the effect of our decay and warmup schedules. While we observe a clear benefit from the decay schedule, the warmup seem to be mostly useful in the object discovery setting, where it helps avoid failure cases of getting stuck in a suboptimal solution (e.g., clustering the image into stripes as opposed to objects).

**Number of training slots** In Figure 13, we show the effect of training with a larger number of slots than necessary. We train both the object discovery and property prediction methods on CLEVR6 with the number of slots we used in our CLEVR10 experiments (note that for object discovery we use an additional slot to be consistent with the baselines). We observe that knowing the precise number of

objects in the dataset is generally not required. Training with more slots may even help in the property prediction experiments and is slightly harmful in the object discovery. Overall, this indicates that the model is rather robust to the number of slots (given enough slots to model each object independently). Using a (rough) upper bound to the number of objects in the dataset seem to be a reasonable selection strategy for the number of slots.

**Soft k-means** Slot Attention can be seen as a generalized version of the soft k-means algorithm [11]. We can reduce Slot Attention to a version of soft k-means with a dot-product scoring function (as opposed to the negative Euclidean distance) by simultaneously replacing the GRU update, all LayerNorm functions and the key/query/value projections with the identity function. Specifically, instead of the GRU update, we simply take the output of Line 8 in Algorithm 1 as the next value for the slots. With these ablations, the model achieves $75.5 \pm 3.8\%$ ARI on CLEVR6, compared to $98.8 \pm 0.3\%$ for the full version of Slot Attention.

Figure 1: Aggregation function variants (Line 8) for object discovery on CLEVR6 (left) and property prediction on CLEVR10 (right).

Figure 2: Ablation on the position embedding for object discovery on CLEVR6 (left) and property prediction on CLEVR10 (right).

Figure 3: Slot initialization variants for object discovery on CLEVR6 (left) and property prediction on CLEVR10 (right).

Figure 4: Choice of softmax axis for object discovery on CLEVR6 (left) and property prediction on CLEVR10 (right).

Figure 5: Slot update function variants for object discovery on CLEVR6 (left) and property prediction on CLEVR10 (right).

Figure 6: Number of attention iterations during training for object discovery on CLEVR6 (left) and property prediction on CLEVR10 (right).

Figure 7: LayerNorm in the Slot Attention Module for object discovery on CLEVR6 (left) and property prediction on CLEVR10 (right).

Figure 8: Optional feedforward MLP for object discovery on CLEVR6 (left) and property prediction on CLEVR10 (right).

Figure 9: Softmax temperature in the Slot Attention Module for object discovery on CLEVR6 (left) and property prediction on CLEVR10 (right).

Figure 10: Offset in the attention maps or the denominator of the weighted mean for object discovery on CLEVR6 (left) and property prediction on CLEVR10 (right).

Figure 11: Learning rate decay for object discovery on CLEVR6 (left) and property prediction on CLEVR10 (right).

Figure 12: Learning rate warmup for object discovery on CLEVR6 (left) and property prediction on CLEVR10 (right).

Figure 13: Number of training slots on CLEVR6 for object discovery (left) and property prediction (right).

## 3 Further Experimental Results

### 3.1 Object Discovery

**Runtime** Experiments on a single V100 GPU with 16GB of RAM with 500k steps and a batch size of 64 ran for approximately 7.5hrs for Tetrominoes, 24hrs for multi-dSprites, and 5 days, 13hrs for CLEVR6 (wall-clock time).

**Qualitative results** In Figure 14, we show qualitative segmentation results for a Slot Attention model trained on the object discovery task. This model is trained on CLEVR6 but uses $K = 11$ instead of the default setting of $K = 7$ slots during both training and testing, while all other settings remain unchanged. In this particular experiment, we trained 5 models using this setting with 5 different random seeds for model parameter initialization. Out of these 5 models, we found that a single model learned the solution of placing the background into a separate slot (which is the one we visualize). The typical solution that a Slot Attention-based model finds (for most random seeds) is to distribute the background equally over all slots, which is the solution we highlight in the main paper. In Figure 14 (bottom two rows), we further show how the model generalizes to scenes with more objects (up to 10) despite being trained on CLEVR6, i.e., on scenes containing a maximum of 6 objects.

### 3.2 Set Prediction

**Runtime** Experiments on a single V100 GPU with 16GB of RAM with 150k steps and a batch size of 512 ran for approximately 2 days and 3hrs for CLEVR (wall-clock time).

**Qualitative results** In Table 1, we show the predictions and attention coefficients of a Slot Attention model on several challenging test examples for the supervised property prediction task. The model was trained with default settings ($T = 3$ attention iterations) and the images are selected by hand to highlight especially difficult cases (e.g., multiple identical objects or many partially overlapping

Figure 14: Visualization of overall reconstructions, alpha masks, and per-slot reconstructions for a Slot Attention model trained on CLEVR6 (i.e., on scenes with a maximum number of 6 objects), but tested on scenes with up to 10 objects, using $K = 11$ slots both at training and at test time, and $T = 3$ iterations at training time and $T = 5$ iterations at test time. We only visualize examples where the objects were successfully clustered after $T = 5$ iterations. For some random slot initializations, clustering results still improve when run for more iterations. We note that this particular model learned to separate the background out into a separate (but random) slot instead of spreading it out evenly over all slots.

objects in one scene). Overall, we can see that the property prediction typically becomes more accurate with more iterations, although the accuracy of the position prediction may decrease. This is not surprising as we only apply the loss at $t = 3$, and generalization to more time steps at test time is not guaranteed. We note that one could alternatively apply the loss at every iteration during training, which has the potential to improve accuracy, but would increase computational cost. We observe that the model appears to handle multiple copies of the same object well (top). On very crowded scenes (middle and bottom), we note that the slots have a harder time segmenting the scene, which can lead to errors in the prediction. However, more iterations seem to sharpen the segmentation which in turns improves predictions.

**Numerical results**  To facilitate comparison with our method, we report the results of Figure 5 of the main paper (left subfigure) in numerical form in Table 2 as well as the performance of DSPN [12] with 10 iterations (as opposed to 30). We note that our approach has generally higher average $AP$ compared to DSPN and lower variance. We remark that the published implementation of DSPN uses a significantly deeper image encoder than our model: ResNet 34 [13] vs. a CNN with 4 layers. Further, we use the same scale for all properties (each coordinate in the prediction vector is in $[0, 1]$), while in DSPN the object-coordinates are rescaled to $[-1, 1]$ and every other property is in $[0, 1]$.

**Results partitioned by number of objects**  Here, we break down the results from Table 2 for the Slot Attention model into separate bins that measure the AP score solely for images with a certain fixed number of objects. This is different from Figure 5 (right subfigure) in the main paper, where we test generalization to more objects at test time. We can observe that the rate of mistakes increases with the number of objects in the scene.

Figure 15: AP score by # of objects.

To analyse to what degree this can be addressed by increasing the number of iterations that are used in the Slot Attention module, we run the same experiment where we increase the number of iterations at test time from 3 to 5 iterations for a model trained with 3 iterations. We can see that increasing the number of iterations significantly improves results for difficult scenes with many objects, whereas this has little effect for scenes with only a small number of objects.

**Step-wise loss & coordinate scaling**  We investigate a variant of our model where we apply the set prediction component and loss at every iteration of the attention mechanism, as opposed to only after the final step. A similar experiment was reported in [12] for the DSPN model. As DSPN uses a different scale for the position coordinates of objects by default, we further compare against a version of our model where we similarly use a different scale. Using a different scale for the object location

Table 1: Example predictions of a Slot Attention model trained with $T=3$ on a challenging example with 4 objects (two of which are identical and partially overlapping) and crowded scenes with 10 objects. We highlight wrong prediction of attributes and distances greater than 0.5.

| Image | Attn. t=1 | Attn. t=2 | Attn. t=3 | Attn. t=4 | Attn. t=5 |

| True Y | Pred. t=1 | Pred. t=2 | Pred. t=3 | Pred. t=4 | Pred. t=5 |
|---|---|---|---|---|---|
| (-2.11, -0.69, 0.70) | (-2.82, -0.19, 0.71), **d=0.87** | (-2.43, -0.22, 0.70), **d=0.56** | (-2.42, -0.55, 0.71), d=0.34 | (-2.41, -0.35, 0.71), d=0.45 | (-2.42, -0.48, 0.71), d=0.36 |
| large blue rubber cylinder | large blue rubber cylinder | large blue rubber cylinder | large blue rubber cylinder | large blue rubber cylinder | large blue rubber cylinder |
| (2.41, -0.82, 0.70) | (2.52, -0.35, 0.66), d=0.48 | (2.57, -0.64, 0.72), d=0.24 | (2.53, -0.83, 0.71), d=0.12 | (2.55, -0.79, 0.70), d=0.14 | (2.54, -0.82, 0.71), d=0.13 |
| large yellow metal cube | large yellow metal cube | large yellow metal cube | large yellow metal cube | large yellow metal cube | large yellow metal cube |
| (-2.57, 1.88, 0.35) | (-2.19, 2.23, 0.37), **d=0.52** | (-2.70, 2.31, 0.34), d=0.45 | (-2.57, 2.35, 0.33), d=0.47 | (-2.58, 2.35, 0.34), d=0.48 | (-2.58, 2.35, 0.34), d=0.48 |
| small purple rubber cylinder | small purple rubber cylinder | small purple rubber cylinder | small purple rubber cylinder | small purple rubber cylinder | small purple rubber cylinder |
| (0.69, -1.51, 0.70) | (0.26, -2.05, 0.67), **d=0.69** | (0.72, -1.47, 0.70), d=0.05 | (0.72, -1.57, 0.69), d=0.07 | (0.71, -1.54, 0.69), d=0.03 | (0.72, -1.55, 0.69), d=0.04 |
| large blue rubber cylinder | large blue rubber cylinder | large blue rubber cylinder | large blue rubber cylinder | large blue rubber cylinder | large blue rubber cylinder |

| Image | Attn. t=1 | Attn. t=2 | Attn. t=3 | Attn. t=4 | Attn. t=5 |

| True Y | Pred. t=1 | Pred. t=2 | Pred. t=3 | Pred. t=4 | Pred. t=5 |
|---|---|---|---|---|---|
| (-2.92, 0.03, 0.70) | (-2.36, -1.24, 0.68), **d=1.39** | (-0.90, 0.35, 0.57), **d=2.05** | (-2.24, 0.16, 0.71), **d=0.70** | (-2.38, -0.04, 0.68), **d=0.55** | (-2.36, 0.03, 0.69), **d=0.56** |
| large green metal cylinder | large green metal cylinder | large green metal cylinder | large green metal cylinder | large green metal cylinder | large green metal cylinder |
| (-1.41, 2.57, 0.35) | (-0.47, 2.41, 0.40), **d=0.95** | (-1.58, 2.11, 0.25), **d=0.50** | (-1.65, 1.99, 0.34), **d=0.63** | (-1.98, 2.10, 0.36), **d=0.74** | (-1.92, 2.24, 0.36), **d=0.61** |
| small gray metal cylinder | small gray metal cylinder | small gray metal cylinder | small gray metal cylinder | small gray metal cylinder | small gray metal cylinder |
| (0.33, 2.72, 0.35) | (0.28, 1.34, 0.38), **d=1.38** | (0.27, 1.06, 0.33), **d=1.66** | (-1.44, 1.56, 0.35), **d=2.11** | (-0.56, 1.16, 0.23), **d=1.80** | (-0.71, 1.24, 0.34), **d=1.81** |
| small blue metal sphere | small blue metal sphere | small blue metal sphere | small blue metal sphere | small blue metal sphere | small blue metal sphere |
| (2.22, -2.28, 0.35) | (2.05, -1.94, 0.37), d=0.38 | (2.28, -2.02, 0.37), d=0.27 | (2.10, -2.03, 0.36), d=0.28 | (2.10, -2.01, 0.36), d=0.30 | (2.09, -2.01, 0.36), d=0.30 |
| small red rubber cube | small red rubber cube | small red rubber cube | small red rubber cube | small red rubber cube | small red rubber cube |
| (1.99, -0.93, 0.70) | (2.03, -0.31, 0.68), **d=0.62** | (1.54, -1.04, 0.72), d=0.47 | (1.90, -0.95, 0.72), d=0.09 | (1.83, -0.90, 0.72), d=0.17 | (1.86, -0.91, 0.72), d=0.14 |
| large blue rubber cube | large blue rubber cube | large blue rubber cube | large blue rubber cube | large blue rubber cube | large blue rubber cube |
| (-1.50, -0.34, 0.35) | (-2.50, 0.14, 0.40), **d=1.11** | (-2.06, 1.66, 0.28), **d=2.08** | (-0.11, 0.72, 0.33), **d=1.76** | (-0.60, 1.13, 0.35), **d=1.72** | (-0.55, 1.04, 0.32), **d=1.68** |
| small blue metal sphere | small **gray** metal **cylinder** | small **gray** metal **cylinder** | small blue metal sphere | small blue metal sphere | small blue metal sphere |
| (1.94, 2.51, 0.35) | (-1.54, 0.85, 0.41), **d=3.86** | (1.15, 2.35, 0.31), **d=0.81** | (1.85, 2.38, 0.37), d=0.23 | (1.76, 2.38, 0.38), d=0.23 | (1.81, 2.34, 0.37), d=0.22 |
| small green metal sphere | **large gray** metal **cylinder** | small green metal sphere | small green metal sphere | small green metal sphere | small green metal sphere |
| (-2.05, -2.99, 0.70) | (-0.45, -1.37, 0.43), **d=2.30** | (-2.53, -2.33, 0.69), **d=0.82** | (-1.53, -2.54, 0.72), **d=0.69** | (-2.30, -2.61, 0.71), d=0.45 | (-2.09, -2.51, 0.70), d=0.48 |
| large gray metal cylinder | large **blue rubber** cylinder | large gray metal cylinder | large gray **rubber** cylinder | large gray metal cylinder | large gray metal cylinder |
| (-0.31, -2.95, 0.70) | (0.10, -2.59, 0.70), **d=0.54** | (-0.25, -2.50, 0.70), d=0.45 | (-0.26, -2.60, 0.69), d=0.35 | (-0.26, -2.69, 0.69), d=0.26 | (-0.29, -2.64, 0.69), d=0.31 |
| large gray rubber cylinder | large gray rubber **cube** | large gray rubber cylinder | large gray rubber cylinder | large gray rubber cylinder | large gray rubber cylinder |
| (1.81, 0.84, 0.35) | (1.16, -1.06, 0.37), **d=2.01** | (0.27, 0.66, -0.17), **d=1.64** | (1.49, 0.32, 0.33), **d=0.62** | (1.40, 0.53, 0.33), **d=0.52** | (1.40, 0.50, 0.33), **d=0.54** |
| small brown rubber sphere | **large blue** rubber **cube** | small brown rubber sphere | small brown rubber sphere | small brown rubber sphere | small brown rubber sphere |

| Image | Attn. t=1 | Attn. t=2 | Attn. t=3 | Attn. t=4 | Attn. t=5 |

| True Y | Pred. t=1 | Pred. t=2 | Pred. t=3 | Pred. t=4 | Pred. t=5 |
|---|---|---|---|---|---|
| (-2.28, 2.76, 0.70) | (-2.40, 2.30, 0.69), d=0.47 | (-1.96, 2.15, 0.67), **d=0.69** | (-2.01, 2.16, 0.66), **d=0.66** | (-1.99, 2.12, 0.66), **d=0.70** | (-1.98, 2.10, 0.66), **d=0.73** |
| large cyan metal sphere | large cyan metal sphere | large cyan metal sphere | large cyan metal sphere | large cyan metal sphere | large cyan metal sphere |
| (0.93, 2.56, 0.35) | (0.00, 1.31, 0.37), **d=1.55** | (0.60, 2.43, 0.33), d=0.35 | (0.85, 2.28, 0.33), d=0.29 | (0.76, 2.39, 0.33), d=0.24 | (0.71, 2.39, 0.33), d=0.28 |
| small purple rubber cube | **large blue metal cylinder** | small purple rubber cube | small purple rubber cube | small purple rubber cube | small purple rubber cube |
| (2.27, -2.44, 0.35) | (-1.14, -1.29, 0.31), **d=3.60** | (-2.61, -2.59, -2.50), **d=5.65** | (2.10, -2.58, 0.35), d=0.22 | (0.81, -1.73, -0.11), **d=1.69** | (2.46, -2.34, 0.38), d=0.22 |
| small purple rubber sphere | **large cyan metal cylinder** | small **green** rubber sphere | small **cyan** rubber sphere | small purple rubber sphere | small purple rubber sphere |
| (-0.70, -2.14, 0.35) | (0.76, -1.79, 0.41), **d=1.50** | (-0.44, -1.97, 0.36), d=0.31 | (-0.17, -1.94, 0.36), **d=0.57** | (-0.31, -1.93, 0.35), d=0.44 | (-0.28, -1.95, 0.35), d=0.46 |
| small yellow metal sphere | small yellow metal sphere | small yellow metal sphere | small yellow metal sphere | small yellow metal sphere | small yellow metal sphere |
| (-0.46, 1.75, 0.70) | (-1.03, 0.70, 0.25), **d=1.27** | (-0.46, 1.61, 0.52), d=0.23 | (-0.45, 2.25, 0.66), **d=0.50** | (-0.83, 2.20, 0.65), **d=0.58** | (-0.70, 2.29, 0.65), **d=0.59** |
| large brown metal sphere | large **yellow** metal sphere | large brown metal sphere | large brown metal sphere | large brown metal sphere | large brown metal sphere |
| (1.14, -0.91, 0.70) | (0.73, -0.68, 0.68), d=0.46 | (0.66, -0.88, 0.71), d=0.48 | (0.74, -0.96, 0.70), d=0.40 | (0.73, -0.91, 0.70), d=0.41 | (0.74, -0.95, 0.69), d=0.40 |
| large green rubber cylinder | large green rubber cylinder | large green rubber cylinder | large green rubber cylinder | large green rubber cylinder | large green rubber cylinder |
| (-2.98, 0.78, 0.70) | (-0.94, 1.14, 0.55), **d=2.08** | (-2.56, 1.25, 0.66), **d=0.63** | (-2.71, 0.56, 0.68), d=0.35 | (-2.73, 1.01, 0.68), d=0.34 | (-2.73, 0.67, 0.68), d=0.27 |
| large brown rubber cylinder | large brown **metal** cylinder | large brown rubber cylinder | large brown rubber cylinder | large brown rubber cylinder | large brown rubber cylinder |
| (-2.51, -2.02, 0.35) | (-2.39, 2.31, 0.57), **d=4.34** | (-2.81, -0.77, 0.36), **d=1.29** | (-2.35, -1.36, 0.33), **d=0.68** | (-2.25, -1.45, 0.32), **d=0.63** | (-2.26, -1.45, 0.32), **d=0.63** |
| small red rubber sphere | large **red** rubber **cube** | small red rubber sphere | small red rubber sphere | small red rubber sphere | small red rubber sphere |
| (1.30, -2.20, 0.35) | (2.27, -2.65, 0.37), **d=1.07** | (2.24, -2.76, 0.36), **d=1.09** | (1.86, -2.38, 0.37), **d=0.58** | (2.03, -2.66, 0.35), **d=0.85** | (1.41, -2.55, 0.34), d=0.36 |
| small cyan rubber cube | small cyan rubber cube | small cyan rubber cube | small cyan rubber cube | small cyan rubber cube | small cyan rubber cube |
| (2.50, 2.80, 0.70) | (2.59, 1.99, 0.72), **d=0.81** | (2.57, 2.72, 0.75), d=0.12 | (2.61, 2.52, 0.75), d=0.30 | (2.60, 2.52, 0.74), d=0.30 | (2.59, 2.51, 0.73), d=0.30 |
| large yellow metal cylinder | large yellow metal cylinder | large yellow metal cylinder | large yellow metal cylinder | large yellow metal cylinder | large yellow metal cylinder |

Table 2: Average Precision at different distance thresholds on CLEVR10 (in %, mean $\pm$ std for 5 seeds). We highlighted the best result for each threshold within confidence intervals.

| | $AP_\infty$ | $AP_1$ | $AP_{0.5}$ | $AP_{0.25}$ | $AP_{0.125}$ |
|---|---|---|---|---|---|
| Slot Attention | **94.3 ± 1.1** | **86.7 ± 1.4** | **56.0 ± 3.6** | **10.8 ± 1.7** | **0.9 ± 0.2** |
| DSPN T=30 | 85.2 ± 4.8 | **81.1 ± 5.2** | **47.4 ± 17.6** | **10.8 ± 9.0** | **0.6 ± 0.7** |
| DSPN T=10 | 72.8 ± 2.3 | 59.2 ± 2.8 | 39.0 ± 4.4 | **12.4 ± 2.5** | **1.3 ± 0.4** |
| Slot MLP | 19.8 ± 1.6 | 1.4 ± 0.3 | 0.3 ± 0.2 | 0.0 ± 0.0 | 0.0 ± 0.0 |

Figure 16: AP scores binned by number of objects in the scene. Difficult scenes that contain many objects require more Slot Attention iterations.

increases its weight in the loss. We observe the effect of the coordinate scale and of computing the loss at each step in Figure 17.

Figure 17: Computing the loss at each iteration generally improves results for both Slot Attention and the DSPN (while increasing the computational cost as well). As expected, re-scaling the coordinate to have a higher weight in the loss, positively impacts the AP at small distance thresholds where the position of objects needs to be predicted more accurately.

A scale of 1 corresponds to our default coordinate normalization of $[0, 1]$, whereas larger scales correspond to a $[0, \mathtt{scale}]$ normalization of the coordinates (or shifted by an arbitrary constant). Overall, we observe that computing the loss at each step in Slot Attention improves the AP score at all distance thresholds as opposed to DSPN, where it is only beneficial at small distance thresholds. We conjecture that this is an optimization issue in DSPN. As expected, increasing the importance of accurately modeling position in the loss impacts the AP positively at smaller distance thresholds, but can otherwise have a negative effect on predicting other object attributes correctly.

## 4 Permutation Invariance and Equivariance

### 4.1 Definitions

Before giving the proof for Proposition 1, we formally define permutation invariance and equivariance.

**Definition 1** (Permutation Invariance). *A function* $f : \mathbb{R}^{M \times D_1} \to \mathbb{R}^{M \times D_2}$ *is permutation invariant if for any arbitrary permutation matrix* $\pi \in \mathbb{R}^{M \times M}$ *it holds that:*

$$f(\pi x) = f(x) \,.$$

**Definition 2** (Permutation Equivariance). *A function* $f : \mathbb{R}^{M \times D_1} \to \mathbb{R}^{M \times D_2}$ *is permutation equivariant if for any arbitrary permutation matrix* $\pi \in \mathbb{R}^{M \times M}$ *it holds that:*

$$f(\pi x) = \pi f(x) \,.$$

### 4.2 Proof

The proof is straightforward and is reported for completeness. We rely on the fact that the sum operation is permutation invariant.

**Linear projections**    As the linear projections are applied independently per slot/input element with shared parameters, they are permutation equivariant.

**Equation 1** The dot product of the attention mechanism (i.e. computing the matrix $M \in \mathbb{R}^{N \times K}$) involves a sum over the feature axis (of dimension $D$) and is therefore permutation equivariant w.r.t. both input and slots. The output of the softmax is also equivariant, as:

$$\texttt{Softmax} \left(\pi_s \cdot \pi_i \cdot M\right)_{(k,l)} = \frac{e^{(\pi_s \cdot \pi_i \cdot M)_{k,l}}}{\sum_s e^{(\pi_s \cdot \pi_i \cdot M)_{k,s}}}$$

$$= \frac{e^{M_{\pi_i(k),\pi_s(l)}}}{\sum_{\pi_s(l)} e^{M_{\pi_i(k),\pi_s(l)}}}$$

$$= \texttt{Softmax} \left(M\right)_{(\pi_i(k),\pi_s(l))},$$

where we indicate with e.g. $\pi_i(k)$ the transformation of the coordinate $k$ with the permutation matrix $\pi_i$. The second equality follows from the fact that the sum is permutation invariant.

**Equation 2** The matrix product in the computation of the `updates` involves a sum over the input elements which makes the operation invariant w.r.t. permutations of the input order (i.e. $\pi_i$) and equivariant w.r.t. the slot order (i.e. $\pi_s$).

**Slot update:** The slot update applies the same network to each slot with shared parameters. Therefore, it is a permutation equivariant operation w.r.t. the slot order.

**Combining all steps:** As all steps in the algorithms are permutation equivariant wrt $\pi_s$, the overall module is permutation equivariant. On the other hand, Equation 2 is permutation invariant w.r.t. to $\pi_i$. Therefore, after the first iteration the algorithm becomes permutation invariant w.r.t. the input order.

## 5 Implementation and Experimental Details

For the Slot Attention module we use a slot feature dimension of $D = D_{\texttt{slots}} = 64$. The GRU has 64-dimensional hidden state and the feedforward block is a MLP with single hidden layer of size 128 and ReLU activation followed by a linear layer.

### 5.1 CNN Encoder

The CNN Encoder used in our experiments is depicted in Table 3 for CLEVR and Table 4 for Tetrominoes and Multi-dSprites. For the property prediction task on CLEVR, we reduce the size of the representation by using strides in the CNN backbone. All convolutional layers use padding SAME and have a bias weight. After this backbone, we add position embeddings (Section 5.2) and then flatten the spatial dimensions. After applying a layer normalization, we finally add $1 \times 1$ convolutions which we implement as a shared MLP applied at each spatial location with one hidden layer of 64 units (32 for Tetrominoes and Multi-dSprites) with ReLU non-linearity followed by a linear layer with output dimension of 64 (32 for Tetrominoes and Multi-dSprites).

Table 3: CNN encoder for the experiments on CLEVR. For the property prediction experiments on CLEVR10 we use stride of 2 on the layers marked with * which decreases the memory footprint.

| Type | Size/Channels | Activation | Comment |
|---|---|---|---|
| Conv $5 \times 5$ | 64 | ReLU | stride: 1 |
| Conv $5 \times 5$ | 64 | ReLU | stride: 1* |
| Conv $5 \times 5$ | 64 | ReLU | stride: 1* |
| Conv $5 \times 5$ | 64 | ReLU | stride: 1 |
| Position Embedding | - | - | See Section 5.2 |
| Flatten | axis: $[0, 1 \times 2, 3]$ | - | flatten x, y pos. |
| Layer Norm | - | - | - |
| MLP (per location) | 64 | ReLU | - |
| MLP (per location) | 64 | - | - |

Table 4: CNN encoder for the experiments on Tetrominoes and Multi-dSprites.

| Type | Size/Channels | Activation | Comment |
|---|---|---|---|
| Conv $5 \times 5$ | 32 | ReLU | stride: 1 |
| Conv $5 \times 5$ | 32 | ReLU | stride: 1 |
| Conv $5 \times 5$ | 32 | ReLU | stride: 1 |
| Conv $5 \times 5$ | 32 | ReLU | stride: 1 |
| Position Embedding | - | - | See Section 5.2 |
| Flatten | axis: $[0, 1 \times 2, 3]$ | - | flatten x, y pos. |
| Layer Norm | - | - | - |
| MLP (per location) | 32 | ReLU | - |
| MLP (per location) | 32 | - | - |

## 5.2 Positional Embedding

As Slot Attention is invariant with respect to the order of the input elements (i.e., it treats the input as a set of vectors, even if it is an image), position information is not directly accessible. In order to give Slot Attention access to position information, we augment input features (CNN feature maps) with positional embeddings as follows: (i) We construct a $W \times H \times 4$ tensor, where $W$ and $H$ are width and height of the CNN feature maps, with a linear gradient $[0, 1]$ in each of the four cardinal directions. In other words, each point on the grid is associated with a 4-dimensional feature vector that encodes its distance (normalized to $[0, 1]$) to the borders of the feature map along each of the four cardinal directions. (ii) We project each feature vector to the same dimensionality as the image feature vectors (i.e., number of feature maps) using a learnable linear map and add the result to the CNN feature maps.

## 5.3 Deconvolutional Slot Decoder

For the object discovery task, our architecture is based on an auto-encoder, where we decode the respresentations produced by Slot Attention with the help of a slot-wise spatial broadcast decoder [14] with shared parameters between slots. Each spatial broadcast decoder produces an output of size width×height×4, where the first 3 output channels denote RGB channels of the reconstructed image and the last output channel denotes a predicted alpha mask, that is later used to recombine individual slot reconstructions into a single image. The overall architecture for used for CLEVR is described in Table 5 and for Tetrominoes and Multi-dSprites in Table 6.

**Spatial broadcast decoder**  The spatial broadcast decoder [14] is applied independently on each slot representation with shared parameters between slots. We first copy the slot representation vector of dimension $D_{\texttt{slots}}$ onto a grid of shape width×height×$D_{\texttt{slots}}$, after which we add a positional embedding (see Section 5.2). Finally, this representation is passed through several de-convolutional layers.

**Slot recombination**  The final output of the spatial broadcast decoder for each slot is of shape width×height×4 (ignoring the slot and batch dimension). We first split the final channels into three RGB channels and an alpha mask channel. We apply a softmax activation function *across* slots on the alpha masks and lastly recombine all individual slot-based reconstructions into a single reconstructed image by multiplying each alpha mask with each respective reconstructed image (per slot) and lastly by performing a sum reduction on this respective output over the slot dimension to arrive at the final reconstructed image. For visualization of the reconstruction masks in a single image, we replace each individual reconstructed image (per slot) with a unique slot-specific color (see, e.g., third column in Figure 14).

## 5.4 Set Prediction Architecture

For the property prediction task, we apply a MLP on each slot (with shared parameters between slots) and train the overall network with the Huber loss following [12]. The Huber loss takes the form of a squared error $0.5x^2$ for values $|x| < 1$ and a linearly increasing error with slope 1 for $|x| \geq 1$. The MLP has one hidden layer with 64 units and ReLU.

Table 5: Deconv-based slot decoder for the experiments on CLEVR.

| Type | Size/Channels | Activation | Comment |
|---|---|---|---|
| Spatial Broadcast | 8×8 | - | - |
| Position Embedding | - | - | See Section 5.2 |
| Conv $5 \times 5$ | 64 | ReLU | stride: 2 |
| Conv $5 \times 5$ | 64 | ReLU | stride: 2 |
| Conv $5 \times 5$ | 64 | ReLU | stride: 2 |
| Conv $5 \times 5$ | 64 | ReLU | stride: 2 |
| Conv $5 \times 5$ | 64 | ReLU | stride: 1 |
| Conv $3 \times 3$ | 4 | - | stride: 1 |
| Split Channels | RGB (3), alpha mask (1) | Softmax (on alpha masks) | - |
| Recombine Slots | - | - | - |

Table 6: Deconv-based slot decoder for the experiments on Tetrominoes and Multi-dSprites.

| Type | Size/Channels | Activation | Comment |
|---|---|---|---|
| Spatial Broadcast | width×height | - | - |
| Position Embedding | - | - | See Section 5.2 |
| Conv $5 \times 5$ | 32 | ReLU | stride: 1 |
| Conv $5 \times 5$ | 32 | ReLU | stride: 1 |
| Conv $5 \times 5$ | 32 | ReLU | stride: 1 |
| Conv $3 \times 3$ | 4 | - | stride: 1 |
| Split Channels | RGB (3), alpha mask (1) | Softmax (on alpha masks) | - |
| Recombine Slots | - | - | - |

The output of this MLP uses a sigmoid activation as we one-hot encode the discrete features and normalize continuous features between $[0, 1]$. The overall network is presented in Table 7

Table 7: MLP for the property prediction experiments.

| Type | Size/Channels | Activation |
|---|---|---|
| MLP (per slot) | 64 | ReLU |
| MLP (per slot) | output size | Sigmoid |

## 5.5   Slot MLP Baseline

For the Slot MLP baseline we predict the slot representation with a MLP as shown in Tables 8 and 9. This module replaces our Slot Attention module and is followed by the same decoder/classifier. Note that we resize images to $16 \times 16$ before flattening them into a single feature vector to reduce the number of parameters in the MLP.

Table 8: Slot MLP architecture for set prediction. This block replaces the Slot Attention module.

| Type | Size/Channels | Activation |
|---|---|---|
| Resize | $16 \times 16$ | - |
| Flatten | - | - |
| MLP | 512 | ReLU |
| MLP | 512 | ReLU |
| MLP | slot size × num slots | - |
| Reshape | [slot size, num slots] | - |

Table 9: Slot MLP architecture for object discovery. This block replaces the Slot Attention module. We use a deeper MLP with more hidden units and a separate slot-wise MLP with shared parameters in this setting, as we found that it significantly improves performance compared to a simpler MLP baseline on the object discovery task.

| Type | Size/Channels | Activation |
|------|---------------|------------|
| Resize | $16 \times 16$ | - |
| Flatten | - | - |
| MLP | 512 | ReLU |
| MLP | 1024 | ReLU |
| MLP | 1024 | ReLU |
| MLP | slot size $\times$ num slots | - |
| Reshape | [slot size, num slots] | - |
| MLP (per slot) | 64 | ReLU |
| MLP (per slot) | 64 | - |

Table 10: Other hyperparameters for all experiments.

(a) Shared hyperparameters.

| Name | Value |
|------|-------|
| `attn`: $\epsilon$ | 1e-08 |
| Adam: $\beta_1$ | 0.9 |
| Adam: $\beta_2$ | 0.999 |
| Adam: $\epsilon$ | 1e-08 |
| Adam: learning rate | 0.0004 |
| Exponential decay | rate 0.5 |
| Slot dim. | 64 |

(b) Hyperparameters for object discovery.

| Name | Value |
|------|-------|
| Warmup iters. | 10K |
| Decay steps | 100K |
| Batch size | 64 |
| Train steps | 500K |

(c) Hyperparameters for property prediction.

| Name | Value |
|------|-------|
| Warmup iters. | 1K |
| Decay steps | 50K |
| Batch size | 512 |
| Train steps | 150K |

## 5.6 Other Hyperparameters

All shared hyperparameters common to each experiments can be found in Table 10a. The hyperparameters specific to the object discovery and property prediction experiments can be found in Tables 10b and 10c respectively.

In both experiments, we use a learning rate warm-up and exponential decay schedules. For the learning rate warm-up, we linearly increase the learning rate from zero to the final learning rate during the first steps of training. For the decay, we decrease the learning rate by multiplying it by an exponentially decreasing decay rate:

$$\text{learning\_rate} * \text{decay\_rate}^{(\text{step}/\text{decay\_steps})}$$

where the decay rate governs how much we decrease the learning rate. See Table 10 for the parameters of the two schedules.

## 5.7 Hyperparameter Optimization

We started with an architecture and a hyperparameter setting close to that of [15]. We tuned hyperparameters on the object discovery task based on the achieved ARI score on a small subset of training images (320) from CLEVR. We only considered 5 values for the learning rate $[1e-4, 4e-4, 2e-4, 4e-5, 1e-5]$ and batch sizes of $[32, 64, 128]$. For property prediction, we took the same learning rate as in object discovery and we computed the AP on a small subset of training images (500). We considered batch sizes of $[64, 128, 512]$ (as we were able to fit larger batches onto a single GPU due to the lower memory footprint of this model).

## 5.8 Datasets

**Set Prediction** We use the CLEVR [16] dataset, which consists of rendered scenes containing multiple objects. Each object has annotations for position $(x, y, z)$ coordinates in $[-3, 3]$), color (8 possible values), shape (3 possible values), material, and size (2 possible values). The number of

objects varies between 3 and 10 and, similarly to [12], we zero-pad the targets so that their number is constant in the batch and add an extra dimension indicating whether labels correspond to true objects or padding. For this task, we use the original version of CLEVR [16] to be consistent with [12] and compare with their reported numbers as well as our best-effort re-implementation. We pre-process the object location to be in $[0, 1]$ and reshape the images to a resolution of $128 \times 128$. Image features (RGB values) are normalized to $[-1, 1]$.

**Object Discovery** For object discovery, we use three of the datasets provided by the Multi-Object Datasets library [17], available at `https://github.com/deepmind/multi_object_datasets`. See the aforementioned repository for a detailed description of the datasets. We use CLEVR (with masks), Multi-dSprites, and Tetrominoes. We split the TFRecords file of the CLEVR (with masks) dataset into multiple shards to allow for faster loading of the dataset from disk. We normalize all image features (RGB values) to $[-1, 1]$. Images in Tetrominoes and Multi-dSprites are of resolution $35 \times 35$ and $64 \times 64$, respectively. For CLEVR (with masks), we perform a center-crop with boundaries $[29, 221]$ (width) and $[64, 256]$ (height), as done in [15], and afterwards resize the cropped images to a resolution of $128 \times 128$. As done in [15], we filter the CLEVR (with masks) dataset to only retain scenes with a maximum number of 6 objects, and we refer to this dataset as CLEVR6, whereas the original dataset is referred to as CLEVR10.

## 5.9 Metrics

**ARI** Following earlier work [15], we use the Adjusted Rand Index (ARI) [18, 19] score to compare the predicted alpha masks produced by our decoder against ground truth instance segmentation masks. ARI is a score that measures clustering similarity, where an ARI score of 1 corresponds to a perfect match and a score of 0 corresponds to chance level. We exclude the background label when computing the ARI score as done in [15]. We use the implementation provided by the Multi-Object Datasets library [17], available at `https://github.com/deepmind/multi_object_datasets`. For a detailed description of the ARI score, see the aforementioned repository.

**Average Precision** We consider the same setup of Zhang et al. [12], where the average precision is computed across all images in the validation set. As the network predicts a confidence for each detection (real objects have target 1, padding objects 0), we first sort the predictions based on their prediction confidence. For each prediction, we then check if in the corresponding ground truth image there was an object with the matching properties. A detection is considered a true positive if the discrete predicted properties (obtained with an argmax) exactly match the ground truth and the position of the predicted object is within a distance threshold of the ground truth. Otherwise, a detection is considered a false positive. We then compute the area under the smoothed precision recall curve at unique recall values as also done in [20]. Ours is a best-effort re-implementation of the AP score as described in [12]. The implementation provided by [12] can be found at `https://github.com/Cyanogenoid/dspn`.

## Footnotes

[1] In this case we aggregate using a weighted sum, as the attention coefficients are normalized over the inputs.