[Reviews · NeurIPS 2020]

Review 1

Summary and Contributions: The paper proposes a simple yet effective module "slot attention" for unsupervised object discovery task. More specifically, the module performs sequential inference on object slots with GRU and attention, of which the implementation closely resembles soft clustering (with roughly one line of change). Compared to other encoder models in the domain, the proposed method has less computational cost, yields good empirical results on unsupervised object discovery and set prediction and generalizes well to unseen scenes. ------------------------------------------------------------ Post-rebuttal update: I am satisfied with the authors' feedback and recommend acceptance.

Strengths: 1. The idea behind the proposed slot attention module is intuitive and simple, in the sense that it can be understood as a learnable and modified version of soft clustering. 2. With empirical results in object discovery and set prediction on different datasets, the authors show that the slot attention module exhibits good out-of-distribution generalization. 3. The computational cost is low computational cost since the operations can mostly be done in a parallel way (and a small number of inference iteration suffices to generate good results).

Weaknesses: 1. The scenes in the experiments are relatively easy: backgrounds and the geometries of the objects are simple. As the slot attention module is essentially an instance segmentor, it may be necessary to see if it can perform instance segmentation for realistic images (e.g. a smaller and customized MS Coco dataset). 2. Although the module seems to have low computational overhead, it will be better if the authors provide quantitative results on the runtime. 3. Unlike some other papers, the model seems not being able to handle a large number of objects (e.g. more than 20) as it may require more iterations to perform reasoning. In addition, it will be good to see some results on runtime to reach certain AP vs. number of objects.

Correctness: Yes.

Clarity: The paper is well-written and easy to follow.

Relation to Prior Work: Yes.

Reproducibility: Yes

Additional Feedback:


Review 2

Summary and Contributions: +++++ Update after rebuttal +++++ The rebuttal has addressed my main questions and I continue to recommend acceptance. +++++ Original review +++++ The paper presents an architectural component for mapping a set of perceptual input vectors to a set of semantically meaningful output entities called slots. The new component, named slot attention, makes use of dot-product attention to ensure order invariance with respect to the input set and order equivariance with respect to the output set. Its effectiveness is demonstrated in the domains of unsupervised object segmentation and set prediction. In both cases, the goal is to train a network which predicts the set of objects in a visual scene, in the first case via an unsupervised clustering approach, in the second via direct supervision. In both domains slot attention either improves performance, reduces resource demands, or both.

Strengths: The theoretical advantages of slot attention are well justified: permutation equivariance has been shown to be critical for set prediction architectures, and previous works have payed a high price for ensuring it, by either predicting elements one-by-one in an autoregressive way, or by introducing expensive inner optimization loops. Slot attention addresses this by ensuring equivariance in set prediction through a relatively low cost component. This is especially topical due to the recent push caused by DETR to view object detection as a set prediction task. The model is evaluated experimentally on two rather distinct domains and compared with the relevant baselines. On the object discovery task, slot attention is either on par with previous work or shows modest improvements. It is argued that the main benefit in this setting is reduced memory consumption and computation time, which is plausible, given the intuition described above. Since the previous models were already designed to do well on these datasets (CLEVR in particular), and no additional inductive biases were added, it is not especially surprising that performance did not improve drastically. The improvement over DSPN on the set prediction domain is substantial and I think it is likely that the method will see widespread use as a result. The reported sanity checks (increasing the number of iteration steps, increasing the number of slots, etc.) indicate that slot attention is reasonably robust.

Weaknesses: The experiment on greyscale CLEVR (Fig. 4) seem to suggest that Slot Attention is capable of distinguishing between multiple objects of identical appearance. This was a major weakness of previous models such as MONet, which tended to cluster by color. Since slot attention does not feature any inductive bias ensuring spatial continuity of objects, it would be good to make this apparent improvement more explicit, for instance, by also running the baselines on greyscale CLEVR, or by exploring the binarized (black/white) version of the sprite dataset. This would help to demonstrate that slot attention also improves performance for object discovery, and not just efficiency. Slot attention makes the assumption that the input is structured as a set. While this holds in many domains, it means that it is not directly applicable to predicting a set from a single input vector, which is a setting that DSPN considered. More generally, since communication between slots only occurs by competing for input vectors, it is not clear that this will be sufficient to model dependencies between output elements in other, perhaps non-visual, set prediction tasks. According to the ablation experiments in the supplementary, the model's performance is somewhat sensitive to learning rate warmup and decay. This is not uncommon, but slightly diminishes the claims that slot attention is simpler and requires fewer hyperparameters, especially since IODINE used a constant learning rate.

Correctness: The claims in the paper and its methodology are generally correct. Regarding memory efficiency, it is stated that slot attention can use a batch size of 64 on V100, while IODINE only used a batch size of 4. However, according to the IODINE paper, this was on a 12 GB GPU, whereas a V100 may have up to 32 GB of memory. It would be good if the exact amount of memory savings could be clarified.

Clarity: The paper is well written and easy to follow.

Relation to Prior Work: The relevant prior work is discussed and the closest competitors are compared against. As far as I know, the paper is also correct in its assertion that DSPN is the only currently published architecture for equivariant set prediction.

Reproducibility: Yes

Additional Feedback: The link for the multi-object dataset in the references is dead, it should be: https://github.com/deepmind/multi_object_datasets A number of works cited via arXiv have been published formally, e.g. Set Transformer (ICML 2019), GENESIS, FSPool, and C-SWM (ICLR 2020).


Review 3

Summary and Contributions: This paper proposes a slot attention mechanism to learn object-centric representations from perceptual input. The attention mechanism can be introduced on top of CNN features to get the slot representations. The slot representations kindly group the separate objects that may facilitate object discovery, set predication, segmentation, etc. Experimental results shows competitive results on unsupervised object discovery and supervised property prediction tasks. ------------------------------------------------------------ Updates: The rebuttal addresses my concerns and I recommend to accept this work.

Strengths: + The object centric problem is important and the proposed idea is quite interesting. + Systematical experimental demonstration. + The writing is good.

Weaknesses: - The proposed methods are only conducted in simple datasets captured in well-controlled environment. This may limit the application of the proposed methods. It's interesting to see the proposed methods' validity on real images. - Experimental results on object discovery are marginally superior to prior arts. For example, on CLEVR6, only 0.x% improvements exist compared to IODINE. I also notice that "learning rate warmup and an exponential decay schedule in the learning rate are used" where IODINE did not use. It's questionable that with the same tricks and polices, which method is better. - Miss descriptions about how to use slot representations to visualize the whole segmentation mask. - In line 231, the statement is a little bit overclaimed that "on CLEVR6, we can use a batch size of up to 64 on a single V100 GPU as opposed to 4 in IODINE." IODINE uses a 12GB GPU but this paper uses a single V100. The V100's memory ranges from 16Gb to 32GB. Without the same memory cost, it's unfair to compare the batch size of two methods.

Correctness: Yes

Clarity: Yes

Relation to Prior Work: Yes

Reproducibility: Yes

Additional Feedback: Although lack experiments on natural images, I think this paper is also valuable to current object discovery and set predication community, and may inspire several applications and improvement.


Review 4

Summary and Contributions: The paper introduces Slot Attention Networks, which are a new class of neural network which act as conduit between low-level perceptual input and high-level, object-like set structured representations. The module learns to iteratively cluster input features into groups which can be thought of as objects. The output is permutation invariant, any object can appear in any slot, which is enabled by choosing initial object states randomly on each run. ------------------------------------------------------------ Updater after rebuttal: Author feedback was reasonable, recommend acceptance.

Strengths: Method is easy to understand, which is appealing, especially compared to a lot of the work in this area. Seems to cut through past work like MONET and IODINE and simplify it quite a bit, which I think is an important contribution. Empirical evaluation seems relatively thorough. Insights of attention normalized over slots (for each input feature) and random slot initializations for permtuation invariance are quite interesting.

Weaknesses: Would be interesting to compare against Soft K-Means, to see if the improvements over Soft K-Means made in your model make a different. I imagine they would, but it'd be satisfying to see. Also, would like to see a dataset with more complicated backgrounds, even e.g. Rooms from the GQN datasets. And even real-world datasets, as done in IODINE. As far I could see there were no examples shown of objects of the same color adjacent to one one another, how does the model fair in such as case? The grayscale example sort of gets at this, but i'd be more interested to see a case where all objects were same color and same material, to see if network is able to segment base on *shape cues only*.

Correctness: Yes.

Clarity: Yes.

Relation to Prior Work: Yes.

Reproducibility: Yes

Additional Feedback: * No mention is made of plans to release code, please release code. * For Hungarian algorithm, what was used as matching distance?

[Author Response · NeurIPS 2020]

We thank the reviewers for their consideration of our paper and their insightful suggestions that will be included in the revision. The consensus appears to be that this is a *"well written"* (R1, R2, R3, R4) paper that introduces a *"simple yet effective module"* (R1) to solve an *"important problem"* (R3) with theoretical advantages that are *"well justified"* (R2). Our approach achieves *"competitive results"* (R3), *"generalizes well"* (R1), and for set prediction tasks *"it is likely that the method will see widespread use"* (R1). We kindly address the reviewers' questions below.

**Memory efficiency compared to IODINE (R2, R3)**

Following our inquiry, the authors of IODINE updated their paper. We both used the same type of hardware: *"V100 GPUs with 16GB of RAM"* (Appendix A.2 in IODINE, arXiv v3).

**Comparisons on realistic images (R1, R3, R4)**

Our method has similar inductive biases to IODINE in the autoencoder setting (for example, we use the same type of decoder), so we believe it would perform similarly on realistic images (Figure 11 in IODINE). The strength of our method rather lies in its simplicity, efficiency, and flexibility.

**Distinguishing between multiple objects of identical appearance (R2, R4)**

As suggested by R2, we ran the object discovery experiment on binarized multi-dSprites to test if we can segment based on shape cues only (also requested by R4). Slot Attention achieves $69.4 \pm 0.9\%$ ARI, whereas the two baselines reported in the IODINE paper achieve $64.8 \pm 17.2\%$ (IODINE) and $68.5 \pm 1.7\%$ (R-NEM). Results on MONet were not reported. This adds further evidence that Slot Attention performs competitively even without being able to rely on color. This experiment along with the discussion will be included in the revision.

**Experimental results on object discovery are marginally superior to prior arts (R2, R3)**

We have comparable inductive biases to IODINE. Our improvements over IODINE mainly concern the simplicity of our approach, its computational cost, and its generality. The use of an attention mechanism in our approach significantly benefits from a learning rate schedule (to prevent early saturation or instability).

**Scalability to more objects and relation between number of objects and iterations (R1)**

Figure 1: AP stratified per number of objects.

To illustrate the limits of Slot Attention and motivate future work on scalability we will add stratified AP results for property prediction on CLEVR using a different number of objects and iterations. An example with threshold 0.5 (where the effect is strongest) is in Figure 1. We clearly see that more complex scenes require more iterations.

Scalability improvements (such as decomposing the image into patches as in the SPACE model) are however orthogonal to our approach and would be interesting to explore in future work.

**Input structured as a set and model dependencies between output elements (R2)**

Concurrent work, Kosiorek et al., "Conditional Set Generation with Transformers" (2020), uses attention to directly communicate between slots, which better addresses the single vector to set example. To model dependencies between output elements one could use a graph neural network as part of the task dependent module.

**Comparison with soft k-means (R4)**

We ran an additional experiment on CLEVR6 where we simultaneously ablated the GRU update, LayerNorm, and the key/query/value projections in the Slot Attention module, which results in a version of soft k-means with a dot-product scoring function (as opposed to Euclidean distance). Using otherwise the same hyperparameters, this model achieves $75.5 \pm 3.8\%$ ARI as opposed to $98.8 \pm 0.3\%$ for Slot Attention. This experiment will be included in the revision.

**Further additions for the final version**

As suggested by the reviewers we will add the runtime details, polish the references, release the code, add details on how reconstruction masks are visualized, and update the link to the datasets.



[Meta-Review · NeurIPS 2020]

This paper proposes a rather simple but useful neural network module called "Slot Attention", which interfaces with perceptual representations such as the output of a convolutional neural network and produces task-dependent abstract representations (slots). The corresponding encoding method enjoys low computational cost and good empirical results on unsupervised object discovery. The authors' response also nicely addressed the main concerns in the original reviews. All reviewers agree that this is an interesting and important contribution to the field of object-centric representation learning.